# FLEx: Personalized Federated Learning for Mixture-of-Experts LLMs via Expert Grafting

## Abstract

Federated instruction tuning of large language models (LLMs) is challenged by significant data heterogeneity across clients, demanding robust personalization. The Mixture of Experts (MoE) architecture, where experts can specialize in distinct data patterns, presents a natural architectural solution to this challenge. The inherent sparsity of the MoE architecture, achieved by selectively activating experts, poses a significant challenge to its integration with federated learning (FL). Conventional FL frameworks, designed for dense models, naively aggregate all expert parameters irrespective of their local activation patterns. This naive approach not only undermines MoE's dynamic sparsity but also risks corrupting the world knowledge within pretrained experts. To address this, we propose FLEx (Federated LLMs with Personalized Experts), a novel framework that leverages pretrained MoE-based LLMs for efficient personalization. By aggregating only the shared non-expert parameters, FLEx significantly reduces communication overhead and preserves the world knowledge stored within the frozen pretrained experts. For personalization, we introduce a novel expert grafting mechanism that leverages dynamic sparsity to construct a client-specific expert from selected components of pretrained experts, tailored to local data. This grafted expert is then fine-tuned locally alongside the gating mechanism. This joint training enables the model to learn when to leverage the shared knowledge from frozen experts and when to employ the personalized one. Evaluations on diverse, non-IID instruction tuning datasets show that FLEx consistently outperforms federated baselines on average, while demonstrating strong knowledge preservation on the knowledge-driven benchmark MMLU. Our code is available at `https://anonymous.4open.science/r/FLEx-8F12`.

## 1 Introduction

The widespread adoption of Large Language Models (LLMs) makes fine-tuning on decentralized user data essential for personalization. Federated Learning (FL) offers a privacy-preserving paradigm for this collaborative training. However, the core challenge of statistical heterogeneity across client data poses a significant challenge to the application of FL to LLMs (Zhang et al., 2024; Ye et al., 2024; Long et al., 2024; Wang et al., 2024b). A straightforward application of standard algorithms like FedAvg often leads to significant performance degradation, where the aggregated global model can underperform even a model trained solely on local data. This is due to destructive interference from conflicting client objectives, causing catastrophic forgetting and undermining both personalization and the model's general capabilities. This highlights an urgent need for an FL framework that can effectively leverage, rather than suffer from, data heterogeneity.

Instead of forcing a single, dense model to handle conflicting data objectives, an alternative paradigm is to employ model architectures that can inherently leverage such diversity. The Mixture of Experts (MoE) (Jacobs et al., 1991) architecture presents a well-suited architectural solution to this problem. By design, MoE models route inputs to specialized "expert" sub-networks via a gating mechanism, allowing different experts to capture distinct data patterns, domains, or linguistic styles (Fedus et al., 2022; Mixtral.AI, 2024). This inherent specialization makes MoE a conceptually ideal candidate for federated settings, where experts could theoretically learn to handle the diverse data distributions from different clients. Crucially, this direction has become particularly practical and promising with the recent emergence of powerful, open-source MoE-based LLMs (DeepSeek-AI, 2025; Kimi et al., 2025; Qwen et al., 2025). The success of these models comes from an architectural tradeoff:

*they achieve superior performance by sparsely activating experts, but at the cost of a massive total parameter count* (e.g., DeepSeek-V3 activates 37B out of total 671B parameters, Kimi-K2 activates 32B out of 1000B parameters).

Conventional FL frameworks, designed for dense models, require aggregating structurally identical parameters from all clients. Since expert activation is sparse and varies across clients based on their local data, all experts must be communicated to meet this structural requirement. This process not only prevents the system from leveraging MoE's dynamic sparsity but also risks corrupting the specialized knowledge within pretrained experts through naive averaging. These key obstacles, particularly the risk of catastrophic forgetting that undermines the critical pretrained foundation of large-scale LLMs, pose a significant challenge to the practical application of large MoE models in personalized, privacy-preserving federated settings.

To address these challenges, we introduce FLEx (Federated LLMs with Personalized Experts), a novel federated learning framework specifically designed for pretrained MoE-based LLMs. Our approach stems from a key insight into the MoE architecture: the dense non-expert parameters (e.g., attention layers) that process all tokens serve as a repository for common knowledge, while the sparsely activated experts are ideal candidates for personalization. This natural division directly informs our decoupling strategy, whereby FLEx builds a shared foundation by exclusively aggregating the dense, common parameters, while simultaneously enabling personalization by keeping the pretrained experts frozen and introducing a novel expert grafting mechanism for clients to construct their own lightweight experts. Our main contributions are summarized as follows:

- We propose FLEx, a novel federated learning framework that addresses the prohibitive communication costs and risk of knowledge corruption inherent in naively applying FL to MoE models. By strategically aggregating only the dense non-expert parameters while keeping the pretrained experts frozen, FLEx significantly diminishes communication overhead and reduces catastrophic forgetting of specialized world knowledge.

- To tackle the core challenge of data heterogeneity, FLEx introduces a novel expert grafting mechanism for effective personalization. Our framework allows each client to construct a lightweight, personalized expert by pruning the frozen, pretrained experts based on local data. A jointly-tuned adaptive gating mechanism then learns to dynamically integrate this personalized expert with the shared knowledge from the base model.

- We conduct extensive evaluations on diverse, non-IID instruction-tuning datasets. The results demonstrate that FLEx markedly outperforms established federated baselines in personalized performance while effectively preserving the general knowledge of the pretrained model, as validated by strong performance on the MMLU benchmark.

## 2 RELATED WORK

### 2.1 FEDERATED LEARNING ON LARGE LANGUAGE MODELS

Federated Learning has gained significant attention as a promising method for training LLMs, addressing challenges such as decentralized data management and privacy concerns (Zhang et al., 2024; Ye et al., 2024; Wang et al., 2024b). Recent surveys (Yao et al., 2024; Ye et al., 2024) have identified important intersections between FL and LLMs, particularly in areas such as efficient foundation model training, federated fine-tuning techniques, and the collaborative potential of FL in advancing LLM development. Notable contributions in this domain include FedIT (Zhang et al., 2024), which demonstrated the utility of FL in instruction tuning tasks for LLMs. Further innovations involve personalized LLM training methods like DualLoRA (Long et al., 2024) and FDLoRA (QI et al., 2024), which combine federated learning with personalized adapter methods. Other approaches such as FRLoRA (Yan et al.), FLoRA (Wang et al., 2024b), and FlexLoRA (Bai et al.) introduce new aggregation strategies designed to optimize low-rank fine-tuning and allow more flexible model updates in federated settings. Additionally, selective aggregation methods, like those presented in Guo et al. (2025), further improve federated LLM training by refining the aggregation process.

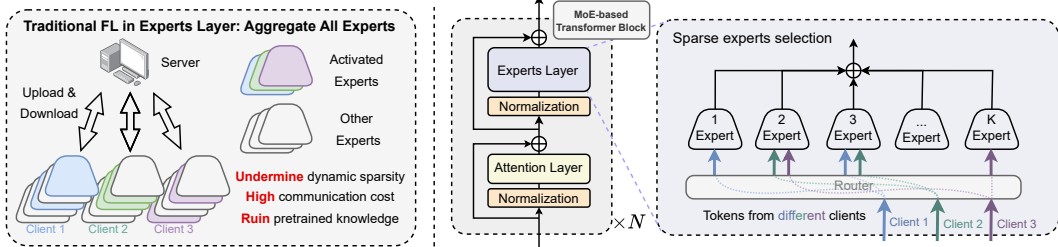

Figure 1: Challenges of applying traditional federated learning to MoE-based LLMs. As shown in the right figure, in an MoE layer, a router sparsely routes tokens from different clients to a selected subset of experts. This dense aggregation scheme conflicts with MoE's sparse activation mechanism, leading to prohibitive communication overhead and undermining the specialized knowledge of the experts.

## 2.2 MIXTURE OF EXPERTS

Mixture of Experts (Jacobs et al., 1991; Fedus et al., 2022) models activate only a subset of experts during inference or training, which significantly reduces computational costs while maintaining high performance. Recently, MoE architectures have gained attention for their scalability and efficiency in large language models (Mixtral.AI, 2024; DeepSeek-AI, 2025; Qwen et al., 2025).

However, existing federated MoE approaches (Zhan et al., 2024; Luo et al., 2025) are primarily designed for dense model architectures. Applying these methods directly to MoE-based LLMs can result in substantial computational and communication overhead. To address this limitation, we propose **FLEx**, a novel framework specifically designed for federated training of MoE-based LLMs.

## 2.3 EXPERT PRUNING FOR LLMS

Model pruning is a widely used technique to improve communication and computational efficiency in deep learning. In the context of MoE-based LLMs, recent pruning methods, such as Wanda (Sun et al., 2024) and SparseGPT (Frantar & Alistarh, 2023), offer effective strategies to reduce the number of model parameters. Additionally, recent studies (Lu et al., 2024; Liu et al., 2024) have demonstrated that many experts are either unimportant or redundant during inference on specific benchmarks. However, in federated learning with MoE-based LLMs, directly pruning certain experts may result in the loss of essential foundational knowledge. To address this, we propose a method for pruning personalized side experts on local data. Furthermore, we integrate the pruned experts back into the original pretrained model to achieve an effective balance between local adaptation and global generalization.

## 3 PRELIMINARY

**Federated Learning Setup**  We consider a federated learning setup with $N$ clients, each client $i \in \{1, \ldots, N\}$ holding a private dataset $\mathcal{D}_i$. In this setup, each client independently trains a model on its local data, while the central server periodically aggregates the updates to improve the global model. A widely adopted approach, the Federated Averaging (FedAvg) algorithm, aggregates the global parameters by computing a weighted average of the local parameters:

$$W^t = \sum_{i=1}^{N} \frac{n_i}{n} W_i^t, \tag{1}$$

where $n = \sum_{i=1}^{N} n_i$ is the total number of training samples across all clients, and $W_i^t$ and $W^t$ are the model weights of $i$-th client and server in $t$-th round, respectively. This approach enables the global model to leverage distributed data while maintaining privacy.

**Mixture of Experts in Large Language Models**  The backbone of an MoE-based LLM is the Transformer architecture. A standard Transformer layer is composed of a self-attention sub-layer

Figure 2: Overview of FLEx framework. The FLEx framework begins by pruning personalized experts for each client using local data. The next step involves injecting personalized knowledge into the MoE layer via a gating mechanism, striking a balance between global knowledge sharing and local adaptation.

and a Feed-Forward Network (FFN) sub-layer. We represent the input to the $l$-th layer as a matrix $\mathbf{H}^{l-1} \in \mathbb{R}^{T \times d_{\text{model}}}$, where $T$ is the sequence length and $d_{\text{model}}$ is the hidden dimension. The layer's operations can be expressed as:

$$\mathbf{U}^l = \text{Self-Att}\left(\text{LayerNorm}(\mathbf{H}^{l-1})\right) + \mathbf{H}^{l-1}, \tag{2}$$

$$\mathbf{H}^l = \text{FFN}\left(\text{LayerNorm}(\mathbf{U}^l)\right) + \mathbf{U}^l, \tag{3}$$

In MoE models, the dense FFN sub-layer is replaced by an MoE layer, which operates on each token's representation individually. This layer consists of $E$ expert networks (each an FFN) and a gating network, or router, that sparsely selects which experts to activate for each token. For the $j$-th token's hidden state $u_j^l$ (a row vector from the output of attention layer $\mathbf{U}^l$), the MoE layer output is computed as:

$$h_j^l = \left(\sum_{i=1}^{E} g_{i,j} \cdot \text{FFN}_i(u_j^l)\right) + u_j^l, \tag{4}$$

where $g_{i,j}$ is the gate value for expert $i$ and token $j$. The gating mechanism employs a Top-K routing strategy. A router network first computes a relevance score $s_{i,j}$ for each expert $i$:

$$s_{\cdot,j} = \text{Softmax}(\text{router}(u_j^l)), \tag{5}$$

The gate values are then determined by selecting the $K$ experts with the highest scores:

$$g_{i,j} = \begin{cases} s_{i,j}, & \text{if } i \in \text{TopK}(\{s_{1,j}, \ldots, s_{E,j}\}, K) \\ 0, & \text{otherwise} \end{cases}, \tag{6}$$

This ensures that only a small fraction of the model's parameters are used for each token, significantly reducing the computational overhead.

Although sparse activation provides significant computational efficiency, it presents a fundamental challenge in the context of federated learning. Standard aggregation methods like FedAvg require the communication of all model parameters. In MoE models, the total number of parameters across all experts is massive, even though only a few are active for any given input. Therefore, directly aggregating all expert layers would result in prohibitive communication costs, making the training process impractical. Furthermore, naively averaging all expert parameters risks corrupting their specialized, pretrained knowledge through destructive interference from heterogeneous client data, potentially leading to catastrophic forgetting.

## 4 UNLOCK THE POWER OF EXPERTS: FLEX

To address the challenges of prohibitive communication costs and knowledge corruption, we introduce FLEx (Federated LLMs with Personalized Experts), a framework designed to decouple shared knowledge from client personalization. The core strategy is to preserve the vast, specialized knowledge within pretrained experts by keeping them frozen locally, while building a robust global model by exclusively aggregating the shared, non-expert parameters. Personalization is then achieved

by constructing a client-specific expert for each experts layer, not from scratch, but through a grafting mechanism. This mechanism repurposes components from the frozen, pretrained experts based on local data. This grafted expert is then integrated into the model and jointly fine-tuned with an adaptive gating mechanism. This joint training process allows the model to learn when to leverage the broad knowledge from the frozen experts and when to activate the personalized expert for client-specific tasks. The overall framework is illustrated in Figure 2.

The following subsections detail the core components of this framework: the selective aggregation strategy (§4.1), the personalized expert grafting mechanism (§4.2), and the adaptive integration process (§4.3).

### 4.1 Selective Aggregation in FLEx

The FLEx framework begins by partitioning the model parameters into two sets: shared non-expert parameters (e.g., self-attention layers, normalization layers) and expert parameters. Following our strategy, only the shared non-expert parameters are aggregated at the server during the federated learning process to form an updated global model.

Crucially, all original pretrained expert layers are frozen on the client side throughout the training. This design directly tackles the challenges outlined previously: it reduces communication costs by excluding the large volume of expert parameters from aggregation, and it preserves the pretrained world knowledge against destructive interference, thereby reducing catastrophic forgetting. This strategy, however, necessitates an effective mechanism for creating personalized components on the client-side. To this end, we introduce our expert grafting process.

### 4.2 Personalized Expert Grafting via Pruning

To address data heterogeneity and enable personalization, FLEx introduces an efficient **expert grafting** mechanism. Instead of training a new expert from scratch, each client constructs a lightweight, personalized expert by leveraging the rich knowledge already present in the frozen pretrained experts. This is accomplished through a targeted pruning process guided by the client's local data.

Specifically, for each MoE layer $l$, each client $i$ identifies the single pretrained expert that best approximates the output of the full expert ensemble on its local data $\mathcal{D}_i$. This selection is framed as an optimization problem that minimizes the reconstruction loss between the output of a single expert and the output of the original multi-expert layer. The index of the selected expert, denoted as $e^*_{i,l}$, for client $i$ at layer $l$ is found by:

$$e^*_{i,l} = \arg\min_{e \in \{1,\ldots,E\}} \sum_{\mathbf{u}^l_j \in \mathcal{D}_i} \left\| \text{FFN}_e(\mathbf{u}^l_j) - \sum_{k=1}^{E} g_{k,j} \cdot \text{FFN}_k(\mathbf{u}^l_j) \right\|^2_F, \tag{7}$$

where $\mathbf{u}^l_j$ is the input to the MoE layer for the $j$-th token, $\text{FFN}_k$ is the $k$-th pretrained expert, $g_{k,j}$ is the original router's gate value, and $\|\cdot\|^2_F$ denotes the squared Frobenius norm. This greedy selection process is performed once at the beginning of training for each client at each MoE layer, efficiently yielding a set of personalized experts $\{\text{FFN}^l_{e^*_{i,l}}\}^L_{l=1}$ that are tailored to the client's data distribution. This collection of grafted experts forms the basis for local adaptation. These selected experts must then be integrated with the frozen pretrained experts, which we achieve through an adaptive gating mechanism detailed next.

### 4.3 Adaptive Integration and Fine-tuning

For each MoE layer $l$, the client's grafted expert is integrated back into the model alongside the frozen pretrained experts. To enable the model to dynamically choose between leveraging shared knowledge and client-specific knowledge, we introduce a new, trainable gating component router$_{e,l}$ dedicated to the personalized expert at that layer. The forward pass of the modified MoE layer (Eq. 4) is updated as:

$$h^l_j = \left( \sum_{k=1}^{E} g_{k,j} \cdot \text{FFN}^l_k(u^l_j) \right) + g^l_{e,j} \cdot \text{FFN}^l_{e^*_{i,l}}(u^l_j) + u^l_j, \tag{8}$$

where the gate value for the personalized expert is computed as:

$$g_{e,j}^l = \text{sigmoid}(\text{router}_{e,l}(u_j^l)). \tag{9}$$

The sigmoid function maps the router's output to a (0, 1) range, acting as a soft, probabilistic gate for the personalized expert. This choice was empirically validated in our ablation studies (see Section 5.3). During local training, updates are restricted to a small subset of parameters: the shared non-expert layers, the personalized expert, and its corresponding gate. All original experts and their corresponding router remain frozen. This joint training allows the model to learn when to rely on the broad knowledge of the pretrained experts and when to activate the personalized expert to handle specific local data patterns.

## 5    EXPERIMENTS

This section presents comprehensive experiments designed to evaluate the effectiveness of FLEx.

### 5.1    EXPERIMENTAL SETUP

**Datasets and Data Partition.**    We utilize the Databricks-dolly-15k[1] dataset, an open-source collection of instruction-following records. To simulate a non-IID data distribution, we partition the dataset by selecting four subtasks with objective, verifiable answers: classification, closed QA, information extraction, and summarization. We conduct our experiments in a cross-silo federated learning setting to evaluate the generalization performance of FLEx across diverse data distributions. This targeted selection ensures that the data on each client corresponds to a specific task category.

**Base Model and Fine-tuning Strategy.**    Our experiments use the Qwen1.5-MoE-A2.7B[2] model as the backbone. For efficient fine-tuning, we apply LoRA (Hu et al., 2022) to the model's trainable layers, configured with a rank of 32 and an alpha of 64. The local model on each client is trained using the Adam optimizer with a learning rate of $4 \times 10^{-5}$ and a batch size of 4. We use `bfloat16` for mixed-precision training and set the maximum input sequence length to 2048. All models are prompted with the Alpaca template. The experiments are conducted on NVIDIA H20 GPUs, each with 96GB of memory.

**Federated Training Protocol and Baselines.**    We implement the federated learning process using the OpenFedLLM framework (Ye et al., 2024). The training proceeds for 100 communication rounds, with each client processing 5 local steps per round. We compare our proposed method, FLEx, against several established federated learning algorithms: FedAvg (McMahan et al., 2017), FedProx (Li et al., 2020), SCAFFOLD (Karimireddy et al., 2020), FedAvgM (Hsu et al., 2019), FedAdagrad (Reddi et al., 2021), FedYogi (Reddi et al., 2021) and FedAdam (Reddi et al., 2021). For all baseline methods, we apply the respective aggregation algorithm to all trainable parameters of the MoE model, including both the expert and non-expert layers. This represents a standard application of these FL algorithms to the MoE architecture.

**Evaluation Metric.**    To measure the instruction-following capability of the trained models, we use the ROUGE-L score as the primary evaluation metric. ROUGE-L is well-suited for our chosen tasks as they have objective, verifiable answers (details in Appendix C.9). For fair comparison, we evaluate the checkpoints from the same communication round across all methods.

### 5.2    MAIN RESULTS ON INSTRUCTION FOLLOWING

**Performance under Pathological Non-IID Data Partition.**    To test the model's ability to handle extreme data heterogeneity, we first evaluate its performance in a pathological non-IID setting where each client's data is confined to a single, distinct task. As shown in Table 1, FLEx achieves the highest average ROUGE-L score, outperforming the next-best federated method, MoE+FedAvgM. Crucially, the inability of most conventional FL algorithms to outperform isolated local training highlights

---

[1]`https://huggingface.co/datasets/databricks/databricks-dolly-15k`
[2]`https://huggingface.co/Qwen/Qwen1.5-MoE-A2.7B`

Table 1: ROUGE-L performance on the Databricks-dolly-15k dataset under a pathological non-IID setting (one task per client). We also report average MMLU scores to assess general knowledge retention. Best and second-best results among FL methods are **bolded** and underlined, respectively. CLF: Classification, CQA: Closed QA, IE: Information Extraction, Summ: Summarization.

| Method | Databricks-dolly-15k | | | | | MMLU |
| | CLF | CQA | IE | Summ | Avg | Avg |
|---|---|---|---|---|---|---|
| Local Training | 51.90 | 34.34 | 40.93 | 40.37 | 41.88 | 44.32 |
| MoE + FedAvg | 51.39 | 34.52 | 37.23 | 41.54 | 41.17 | 43.91 |
| MoE + FedAvgM | 51.69 | 36.72 | 38.54 | **42.54** | 42.37 | 40.13 |
| MoE + FedAdam | 49.77 | 35.88 | 37.45 | 41.11 | 41.05 | 42.57 |
| MoE + FedAdagrad | 50.09 | 36.38 | 38.71 | 42.24 | 41.85 | 47.06 |
| MoE + SCAFFOLD | 51.20 | 34.45 | 37.93 | 41.75 | 41.33 | 45.01 |
| MoE + FedProx | 51.55 | 34.71 | 35.07 | 41.71 | 40.76 | 44.79 |
| MoE + FedYogi | 49.73 | 36.67 | 36.27 | 41.36 | 41.00 | 44.05 |
| **FLEx (Ours)** | **52.10** | **38.23** | **40.13** | 42.09 | **43.13** | **49.74** |

the inherent challenge of forging a single global model from diverging client objectives in highly heterogeneous environments.

To further investigate FLEx's effectiveness, we evaluated the general knowledge preservation of the personalized models using the MMLU(Wang et al., 2024a) benchmark. FLEx achieves the highest MMLU score among the compared methods, suggesting it effectively preserves the foundational knowledge of the base model. We attribute this to our design of freezing the pretrained, shared experts, which protects them from conflicting and potentially catastrophic updates from specialized clients. This approach enables personalized experts to specialize in local tasks without diminishing the model's core capabilities.

Table 2: Average ROUGE-L performance on Databricks-dolly-15k under a Dirichlet distribution with $\alpha = 0.1$ (higher heterogeneity) and $\alpha = 1.0$ (lower heterogeneity). Best and second-best results are **bolded** and underlined.

| Method | Databricks-dolly-15k | |
| | alpha=0.1 | alpha=1.0 |
|---|---|---|
| Local Training | 35.18 | 36.88 |
| MoE + FedAvg | 35.29 | 37.51 |
| MoE + FedAvgM | **37.41** | 35.94 |
| MoE + FedAdam | 34.69 | 35.79 |
| MoE + FedAdagrad | 36.84 | 36.84 |
| MoE + SCAFFOLD | 34.08 | 36.91 |
| MoE + FedProx | 35.05 | 36.68 |
| MoE + FedYogi | 35.26 | 36.41 |
| **FLEx (Ours)** | 37.30 | **37.54** |

**Performance under Dirichlet-based Non-IID Setting.** Finally, we simulate more realistic, yet still challenging, non-IID scenarios by partitioning the Databricks-dolly-15k dataset among 10 clients using a Dirichlet distribution. Table 2 shows that FLEx remains highly competitive. It outperforms the other methods for $\alpha = 1.0$, a setting with moderate heterogeneity, and the second-best for $\alpha = 0.1$, a setting with higher heterogeneity. While the performance gap between methods narrows as data distributions become more balanced, FLEx's consistent leading performance highlights its adaptability. This robust performance can be attributed to its effective personalization, which enables the model to generalize across various degrees of data heterogeneity, not just in extreme cases.

**Helpfulness and Harmlessness Evaluation.** We assessed open-ended generation quality using the Vicuna benchmark in an IID setting with 20 clients. As shown in Figure 3, FLEx achieves the

highest scores for both Helpfulness (6.360) and Harmlessness (7.993). The significant improvement in harmlessness is particularly noteworthy, suggesting that FLEx's personalized approach is better at capturing the nuances required to generate high-quality, safe, and useful responses, outperforming both baselines and the original pre-trained model.

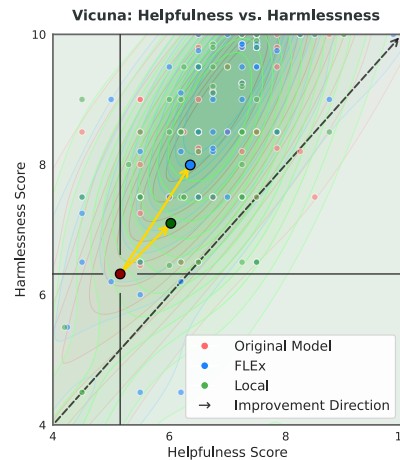

| Method | Vicuna | |
| --- | --- | --- |
| | Helpfulness | Harmlessness |
| Base Model | 5.145 | 6.319 |
| Local Training | 6.026 | 7.096 |
| MoE + Fedavg | 6.159 | 7.661 |
| MoE + FedAvgM | 6.043 | 7.527 |
| MoE + FedAdam | 6.267 | 7.577 |
| MoE + FedAdagrad | 6.314 | 7.780 |
| MoE + SCAFFOLD | 5.908 | 7.248 |
| MoE + FedProx | 5.949 | 7.475 |
| MoE + FedYogi | 5.964 | 7.368 |
| **FLEx (Ours)** | **6.360** | **7.993** |

Figure 3: Evaluation of Helpfulness and Harmlessness on the Vicuna benchmark. Left: Score distributions for the base model, local training, and FLEx. Larger markers indicate average scores. Right: Average scores for all evaluated methods. FLEx achieves the highest scores on both metrics, with a notable improvement in harmlessness.

## 5.3 ABLATION AND COMPONENT ANALYSIS

**Ablation Study on Core Components.** We conducted an ablation study on the Databricks-dolly-15k dataset under a pathological non-IID setting to evaluate the contributions of our two core components: personalized expert grafting and the adaptive gate. As shown in Table 3, naively grafting an expert without our adaptive gate leads to a catastrophic performance collapse, as shown in settings (b) and (c). This occurs because naively inserting an expert without the adaptive gate destabilizes the model's internal dynamics. The gate is crucial for learning to appropriately scale and integrate the expert's contributions. Without it, the raw outputs from the grafted expert can overwhelm or corrupt the knowledge pathways of the frozen base model.

Introducing the adaptive gate, even with a generic expert (d), recovers performance and surpasses the FedAvg baseline. This result highlights the gate's critical role in dynamically merging the knowledge from the frozen and grafted experts. Finally, our complete FLEx model (e), which combines the adaptive gate with a personalized expert, achieves the best results. The improvement from (d) to (e) demonstrates the distinct benefit of tailoring the grafted expert to a client's local data. These results confirm that both components are important: the adaptive gate enables effective knowledge integration, and personalization enhances local adaptation.

Table 3: Ablation study on the core components of FLEx. We analyze the effects of the expert grafting criterion (personalized vs. generic) and the adaptive gating mechanism.

| Model Configuration | ROUGE-L |
| --- | --- |
| (a) Baseline (FedAvg, no FLEx components) | 41.17 |
| *— Naively Integrating a Grafted Expert —* | |
| (b) Generic Grafting w/o Adaptive Gate | 5.16 |
| (c) Personalized Grafting w/o Adaptive Gate | 7.06 |
| *— Integrating via Adaptive Gating —* | |
| (d) Full FLEx w/ Generic Grafting | 42.67 |
| (e) Full FLEx w/ Personalized Grafting **(Ours)** | **43.14** |

**Impact of Training and Communication Strategy.** To validate our architectural choices, we conduct an ablation study evaluating the impact of the training strategy (which experts to train) and the communication strategy (which parameters to share). The results in Table 4 lead to two key observations. First, since each client's experts specialize in distinct local data distributions, naively aggregating all client experts induces destructive interference, which degrades overall performance (row 1 vs. 3 and 2 vs. 4). In contrast, sharing only the common non-expert parameters is more effective, as they capture shared knowledge without this specialized interference. Second, under identical communication costs (rows 3-5), FLEx's personalized expert grafting (row 5) substantially outperforms training all experts (row 3) or only the most activated one (row 4). These findings confirm that FLEx's strength lies in its effective combination of targeted personalization and communication efficiency, rather than simply concentrating training resources.

Table 4: Performance comparison of different strategies for training and sharing MoE modules on Databricks-dolly-15k (pathological non-IID). Our method (FLEx) achieves the highest performance with the lowest communication cost. CP: Communication Parameters, TP: Trainable Parameters.

| Strategy | | Databricks-dolly-15k | | | | | Cost (%) | |
|---|---|---|---|---|---|---|---|---|
| Trained Experts | Shared Parameters | CLF | CQA | IE | Summ | Avg | CP ↓ | TP ↓ |
| All | Non-Experts + All Experts | 51.39 | 34.52 | 37.23 | 41.54 | 41.17 | 3.4172 | 3.4172 |
| Activated | Non-Experts + All Experts | 50.83 | 34.34 | 36.26 | 41.59 | 40.75 | 0.1327 | **0.1327** |
| All | Non-Experts Only | 51.10 | 32.51 | **41.39** | 40.60 | 41.40 | **0.0672** | 3.4172 |
| Activated | Non-Experts Only | 51.48 | 34.77 | 37.23 | 41.74 | 41.30 | **0.0672** | **0.1327** |
| **Personalized** | **Non-Experts Only (FLEx)** | **52.10** | **38.23** | 40.13 | **42.09** | **43.13** | **0.0672** | **0.1327** |

**Ablation on the Gating Activation Function.** We investigated the choice of activation function for the grafted expert's gating mechanism, as defined in Eq. (9). We compared our default sigmoid function against two common alternatives: ReLU and Tanh. The results, summarized in Table 5, demonstrate that the sigmoid-based gate consistently and significantly outperforms the other functions. A detailed discussion on the rationale behind this performance difference is shown in Appendix C.3.

Table 5: Experiments under different side expert aggregation strategies. ROUGE-L performance on the Databricks-dolly-15k dataset with Qwen1.5-MoE-A2.7B under pathological non-IID scenario.

| | Databricks-dolly-15k | | | | |
|---|---|---|---|---|---|
| Method | CLF | CQA | IE | Summ | Avg |
| ReLU | 51.16 | 33.02 | 37.51 | 41.78 | 40.86 |
| Tanh | 41.40 | 20.40 | 31.34 | 37.88 | 32.75 |
| Sigmoid | **52.10** | **38.23** | **40.13** | **42.09** | **43.13** |

**Compatibility with PEFT.** FLEx is compatible with Parameter-Efficient Fine-Tuning (PEFT) methods like LoRA. By applying LoRA or other PEFT methods to the trainable components in FLEx, only a small set of adapter parameters are updated instead of the full weights. This strategy leverages dynamic sparsity to further reduce both computational and communication overhead in FL.

## 6  CONCLUSION

In this work, we introduced FLEx, a novel federated learning framework designed to effectively harness the power of pretrained MoE-based LLMs in settings with heterogeneous data. Our approach addresses the critical challenges of catastrophic forgetting and prohibitive communication costs by selectively aggregating only the shared, dense parameters while freezing the pretrained experts to preserve their specialized knowledge. For personalization, we proposed a novel expert grafting mechanism that allows clients to construct lightweight, customized experts tailored to their local data, which are then dynamically integrated with the frozen experts via a jointly trained gating mechanism. Extensive evaluations demonstrate that FLEx significantly outperforms established federated baselines on personalization tasks across diverse non-IID datasets, while simultaneously preserving the model's general knowledge as validated by strong performance on the MMLU benchmark.

## REPRODUCIBILITY STATEMENT

To support the reproducibility of our research, we detail our experimental setup and hyperparameters in Section 5.1. All datasets used are publicly available. Furthermore, our source code will be publicly released upon publication.

## ETHICS STATEMENT

This research was conducted ethically. We used only publicly available datasets, ensuring that no sensitive or private data was involved. We have also considered the broader impact of this work and foresee no negative societal consequences.

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

SUPPLEMENTARY ORGANIZATION:

## A  ALGORITHM FRAMEWORK

We now present a detailed overview of our FLEx (**F**ederated **L**LMs with Personalized **Ex**perts) framework. The cornerstone of FLEx lies in this separation: **the personalized experts and routers remain strictly local**, preserving client personalization. In contrast, **the non-expert layers are aggregated globally**, enabling shared knowledge to propagate across the federation. This design allows FLEx to effectively balance powerful personalization with collaborative learning. The process begins with a one-time personalized expert selection for each client (detailed in Algorithm 2). Subsequently, the training proceeds in iterative rounds, as outlined in Algorithm 1. Each round consists of two primary steps:

1. **Local Training:** Each client finetunes its model by training all non-expert layers and its personalized subset of experts within the MoE layers. The client-specific router is also updated during this step.

2. **Global Aggregation:** After local training, clients transmit **only the parameters of the non-expert layers** to the server. The server performs federated averaging (FedAvg) on these parameters and sends the aggregated global non-expert layers back to the clients, who integrate them into their local models.

---

**Algorithm 1** FLEx: **F**ederated **L**LMs with Personalized **Ex**perts

---

1: **Input:** Pre-trained MoE model, client set $\mathcal{C}$
2: **for** each client $i \in \mathcal{C}$ **do**
3:     Perform personalized expert selection (Algorithm 2)
4:     Update local model with personalized MoE layer
5: **end for**
6: **for** each global iteration **do**
7:     **for** each client $i \in \mathcal{C}$ **do**
8:         Send parameters of non-expert layers to server
9:         Keep personalized MoE layer and routers locally
10:     **end for**
11:     Aggregate parameters of non-expert layers
12:     Send aggregated non-expert layers to clients and integrate into personalized models locally
13: **end for**
14: **Output:** Federated model with personalized experts on each client

---

Algorithm 2 provides the detailed procedure for the initial personalized expert selection. Specifically, we employ a reconstruction loss strategy to prune the original MoE layers down to a tailored subset of experts uniquely optimized for each client. The selected subset effectively minimizes the discrepancy between the outputs of the full expert set and those produced by the personalized subset, thus ensuring strong local performance.

---

**Algorithm 2** Personalized Expert Selection

---

1: **Input:** Local dataset $\mathcal{D}_i$ for client $i$, pre-trained MoE weights, number of experts $K$, size of pruned subset $n$
2: Initialize local model with pre-trained MoE weights
3: **for** each MoE layer $l$ independently **do**
4:     **for** each input $x \in \mathcal{D}_i$ **do**
5:         Compute the outputs of all experts: $F(x)$
6:     **end for**
7:     Select personalized expert subset $e^*_{i,l}$ by solving:

$$e^*_{i,l} = \arg\min_{e \in \{1,\dots,E\}} \sum_{\mathbf{u}^l_j \in \mathcal{D}_i} \left\| \text{FFN}_e(\mathbf{u}^l_j) - \sum_{k=1}^{E} g_{k,j} \cdot \text{FFN}_k(\mathbf{u}^l_j) \right\|^2_F,$$

8:     Prune the MoE layer by retaining only experts in $e^*_{i,l}$
9: **end for**
10: **Output:** Personalized expert subsets and updated MoE layers

---

## B  Discussion

Our method retains the pre-trained weights in the MoE layer while adding a personalized side expert. This design effectively preserves pre-trained knowledge, leverages personalized experts to enhance client-specific capabilities, benefits from shared global knowledge aggregation, and significantly reduces communication overhead.

**Limitation.**  A key limitation of FLEx is the restriction to grafting only a single personalized expert for each client per layer. As discussed in Appendix C.2, while a straightforward extension would be to greedily select multiple experts, this approach would introduce substantial computational costs without guaranteeing an optimal combination. Therefore, developing more effective pruning algorithms that can efficiently identify a globally optimal set of personalized experts is a direction for future research.

**Further Comparsion with Previous Works.**  It is worth mentioning that Mei et al. (2024) proposed a federated learning approach specifically targeting frameworks utilizing Switch Transformers to address data heterogeneity. However, their approach involves fine-tuning all MoE layers, leading to significant forgetting from the pretrained models. This forgetting issue is particularly concerning given that current decoder-only large language models are pretrained on billions of tokens, making the preservation of pretrained knowledge an important task in federated learning. In contrast, our proposed method maintains the integrity of the original pretrained model parameters by introducing a specialized side-expert specifically designed to handle data heterogeneity. Our approach effectively mitigates the forgetting issue inherent in methods that fine-tune all MoE layers. Furthermore, we validate the applicability and efficacy of our federated learning approach through extensive experiments on billion-token pretrained, instruction-tuned large language models, including Qwen and DeepSeek.

**Efficiency Analysis.**  Although our method introduces an additional personalized expert, it significantly reduces training overhead compared to conventional fine-tuning by restricting backpropagation exclusively to this new expert and its adaptive gate. Moreover, the impact on inference latency is negligible, as the grafted expert can be processed in parallel with the frozen pretrained experts, capitalizing on the inherent sparsity of the MoE architecture.

## C  Further Experiments

### C.1  Evaluation Across Different Model Architectures

In this section, we validate the robustness of FLEx across different model architectures. Specifically, we conduct experiments using the Databricks-dolly-15k dataset in combination with an alternative MoE-based model, DeepSeek-MoE-16B-Base[3]. We assess performance under a pathological non-IID scenario, with the results summarized in Table 6. From the table, we observe significant improvements across most evaluated tasks, with only a slight reduction in classification performance. When considering the average performance gains, our proposed method demonstrates a relatively substantial improvement (32.66) compared to the closest competitor, MoE+FedYogi (30.85). These results confirm that our approach maintains effectiveness and consistently delivers enhanced performance across varying model architectures.

### C.2  Analysis on the Number of Personalized Experts

In the FLEx framework, each client finetunes a small subset of personalized experts while keeping the original, pre-trained experts active. The output of the new personalized expert is integrated with that of the original experts to generate the final output, as detailed in Equation (9).

To determine the optimal number of experts to personalize, we conducted an experiment varying this number from one to three. The results, summarized in Table 7, show that pruning additional experts yields only marginal performance gains while incurring an exponential increase in computational cost.

---

[3]https://huggingface.co/deepseek-ai/deepseek-moe-16b-base

Table 6: Experiments under another MoE-based LLM, DeepSeek-MoE-16B-Base. The following table shown Rouge-L performance comparison of different federated learning strategies on the Databricks-dolly-15k dataset under a pathological non-IID scenario. The best result over federated method is **bolded** and the second-best result is underlined. CLF: Classification, CQA: Closed Question Answering, IE: Information Extraction, Summ: Summarization

| Method | Databricks-dolly-15k | | | | |
| | CLF | CQA | IE | Summ | Avg |
|---|---|---|---|---|---|
| Local Training | 51.71 | 14.17 | 17.70 | 37.15 | 30.18 |
| MoE + FedAvg | 48.33 | 14.74 | 17.89 | 31.92 | 28.97 |
| MoE + FedAvgM | 48.25 | 14.21 | 17.99 | 32.55 | 28.25 |
| MoE + FedAdam | 49.96 | 19.15 | 20.30 | 33.83 | 30.81 |
| MoE + FedAdagrad | 50.10 | 18.75 | 20.83 | 32.91 | 30.64 |
| MoE + SCAFFOLD | 49.35 | 14.34 | 17.41 | 31.30 | 28.10 |
| MoE + FedProx | 49.35 | 14.55 | 17.50 | 32.17 | 28.39 |
| MoE + FedYogi | **50.63** | 18.42 | 20.70 | 33.66 | 30.85 |
| **FLEx (Ours)** | 50.33 | **20.11** | **21.29** | **38.93** | **32.66** |

Table 7: Performance and computational cost comparison for finetuning a varying number of experts per layer. The experiment is conducted on the Databricks-dolly-15k dataset under the pathological non-IID setting. While performance gains are marginal, the required finetuning time increases exponentially.

| Method | CLF | CQA | IE | Summ | Avg | Pruning Time (s) |
|---|---|---|---|---|---|---|
| FLEx (1 Expert) | 52.10 | 38.23 | 40.13 | 42.09 | 43.13 | 58 |
| FLEx (2 Experts) | 51.98 | 38.07 | 41.26 | 42.16 | 43.36 | 3,246 |
| FLEx (3 Experts) | 52.24 | 37.83 | 41.12 | 42.81 | 43.50 | 87,091 |

By personalizing a single expert, FLEx effectively captures client-specific data patterns without imposing prohibitive computational demands. This makes our method practical for real-world federated learning scenarios, where clients typically operate with limited computational resources.

## C.3 DISCUSSION ON THE GATING ACTIVATION FUNCTION

Our empirical results in Section 5.3 highlight the superior performance of the sigmoid activation for the personalized expert's gating mechanism. We attribute this to the inherent properties of the sigmoid function. Its output, bounded within the $(0, 1)$ range, naturally serves as a well-calibrated, probabilistic soft switch. This allows the model to learn a smooth and interpretable weighting for engaging the personalized expert based on the input token.

In contrast, the alternative functions lack this intuitive characteristic. Specifically, the ReLU-based weighting for side expert $g_{e,j}^l$ can be expressed as:

$$g_{e,j}^l = \text{ReLU}(\text{router}_{e,l}(u_j^l)). \tag{10}$$

Similarly, for the Tanh-based aggregation, the weight $g_e^t$ is calculated as:

$$g_{e,j}^l = \text{Tanh}(\text{router}_{e,l}(u_j^l)). \tag{11}$$

The unbounded nature of the ReLU function's output ($[0, \infty)$) can introduce training instability when used as a gate, as it does not represent a normalized weight. Similarly, the Tanh function's output in the $(-1, 1)$ range is less suitable for a gating mechanism, as negative values do not have a clear interpretation for modulating an expert's contribution. Therefore, the sigmoid function provides an inductive bias that is better aligned with the task of dynamically blending a specialized expert with a set of generalist pretrained experts.

## C.4 COMPARISON WITH TRADITIONAL FEDERATED LEARNING

Additionally, we compare our method with traditional federated learning algorithms that treat MoE as dense models from two perspectives: performance and communication cost. The experiment is conducted using the Qwen1.5-MoE-A2.7B model on the Databricks-dolly-15k dataset under a challenging pathological non-IID scenario. As illustrated in Figure 4, our approach achieves comparable performance with significantly lower communication overhead.

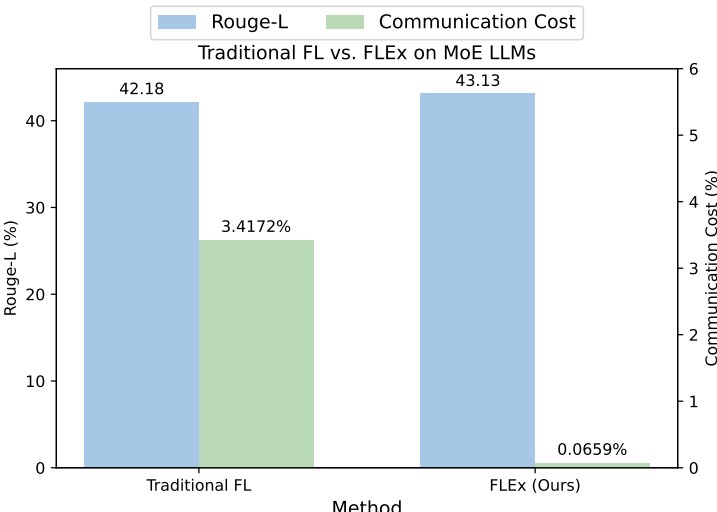

Figure 4: Our method significantly reduces communication overhead while simultaneously enhancing performance, a stark contrast to traditional federated learning approaches that treat MoE as dense models, leading to prohibitive communication costs.

## C.5 ROBUSTNESS ACROSS DIVERSE KNOWLEDGE DOMAINS

To investigate whether FLEx's advantages extend from task-based heterogeneity to domain-based heterogeneity, we conducted experiments on three domain-specific datasets: Alpaca-gpt4 (general), Finance-Alpaca (finance), and MedAlpaca (medical). The results in Table 8 further validate our initial findings. FLEx consistently outperforms all baselines across every domain, achieving the highest average ROUGE-L score. This demonstrates that our method is not only effective at managing different instruction-following tasks but is also robust in federated environments where clients represent distinct, specialized knowledge domains.

Table 8: ROUGE-L performance on domain-specific datasets (Alpaca-gpt4, Finance-Alpaca, and MedAlpaca) under a pathological non-IID setting. Best and second-best results among FL methods are **bolded** and underlined.

| Method | Alpaca-gpt4 | Finance-Alpaca | MedAlpaca | Avg |
|---|---|---|---|---|
| Local Training | 31.31 | 28.64 | 30.93 | 30.29 |
| MoE + FedAvg | 30.11 | 28.17 | 30.06 | 29.44 |
| MoE + FedAvgM | 28.90 | 27.18 | 28.78 | 28.28 |
| MoE + FedAdam | 29.23 | 28.70 | 31.18 | 27.70 |
| MoE + FedAdagrad | 30.24 | 28.79 | 30.60 | 29.87 |
| MoE + SCAFFOLD | 29.96 | 28.38 | 28.92 | 29.08 |
| MoE + FedProx | 30.05 | 28.60 | 30.01 | 29.55 |
| MoE + FedYogi | 29.73 | 28.42 | 30.51 | 29.55 |
| **FLEx (Ours)** | **31.54** | **29.89** | **31.31** | **30.91** |

## C.6 EXPERIMENTS ON EXPERT LOAD BALANCING

We conducted experiments to evaluate the activation frequency of experts within the `Qwen1.5-MoE-A2.7B` model using the C4 dataset[4] across different methods. Figure 5 illustrates the activation count for individual experts. Specifically, the original model and the FedAvg method tend to produce uneven workloads, causing certain experts (e.g., expert 31 and 43), to become significantly overloaded. Specifically, the original model exhibits a standard deviation of 1859.80, while FedAvg demonstrates a slightly higher imbalance with a standard deviation of 1906.85. In contrast, our proposed FLEx method significantly improves load balancing among experts, achieving a notably lower standard deviation of 1259.88. This analysis demonstrates that our method achieves a more balanced workload distribution across different experts.

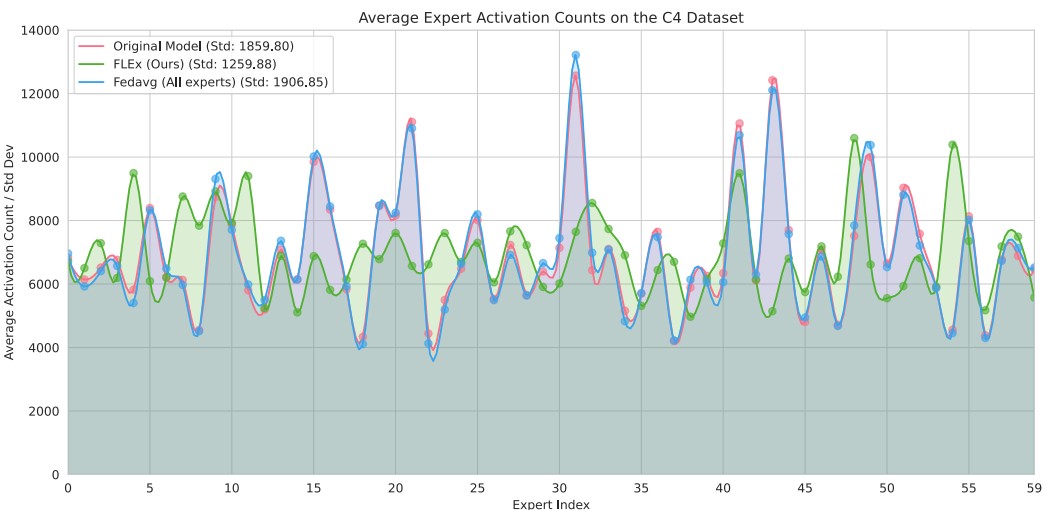

Figure 5: Activation counts of experts in the `Qwen1.5-MoE-A2.7B` model evaluated on the C4 dataset.

## C.7 DATA DISTRIBUTION FOR NON-IID

In this paper, we explore different scenarios for data distribution, specifically focusing on pathological non-IID and Dirichlet non-IID settings. As illustrated in Figure 6, the pathological non-IID scenario assigns four distinct tasks to four separate clients, resulting in an extreme non-IID environment. This scenario significantly complicates the aggregation of models across clients. The Dirichlet non-IID distribution offers a compromise between the pathological non-IID and IID distributions. Specifically, a lower value of the hyperparameter $\alpha$ indicates a higher degree of non-IID distribution among clients. Unless explicitly stated otherwise, this paper defaults to the pathological non-IID distribution.

## C.8 OVERVIEW OF HYPERPARAMETERS

In this section, we present the hyperparameters used for the models `Qwen1.5-MoE-A2.7B` and `DeepSeek-MoE-16B-Base`. Detailed hyperparameter configurations are summarized in Table 9.

## C.9 THE ROUGE-L METRIC

ROUGE-L (Recall-Oriented Understudy for Gisting Evaluation - Longest Common Subsequence) is a metric that evaluates a candidate text by measuring its longest common subsequence (LCS). An LCS is the longest sequence of words that is shared between two texts, maintaining their relative order but not requiring that the words be contiguous.

---

[4]https://huggingface.co/datasets/allenai/c4

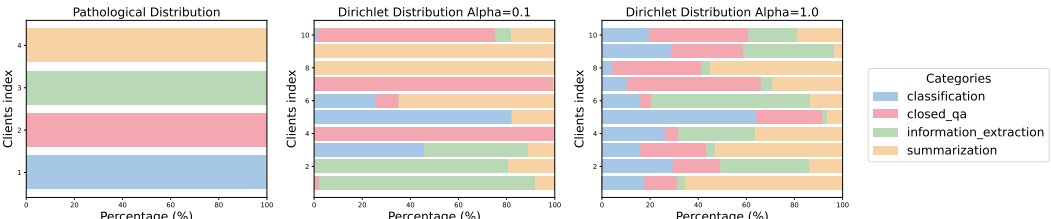

Figure 6: Distribution of instruction types across different clients in federated learning under various settings. The first panel shows the pathological distribution, where each client has a highly imbalanced data distribution. The second panel illustrates the Dirichlet distribution with $\alpha$=0.1, resulting in a more skewed distribution across clients. The third panel presents the Dirichlet distribution with $\alpha$=1.0, demonstrating a more uniform distribution of categories (classification, closed question answering, information extraction, and summarization) across clients.

| Model Name | Qwen1.5-MoE-A2.7B | DeepSeek-MoE-16B-Base |
|---|---|---|
| # Params | 14.3B | 16.4B |
| # Layers | 24 | 28 |
| Hidden Size | 2048 | 2048 |
| # Attn Heads | 16 | 16 |
| # Shared Experts | 1 | 2 |
| # Routed Experts | 60 (4 activated) | 64 (6 activated) |
| Relative Expert Size | 0.25 | 0.25 |
| Sequence Length | 2048 | 2048 |
| Learning Rate | 4e-5 | 4e-5 |

Table 9: Model hyperparameter configurations.

The metric is calculated using LCS-based recall ($R_{lcs}$), precision ($P_{lcs}$), and their F-score ($F_{lcs}$). Given a reference text $X$ of length $m$ and a candidate text $Y$ of length $n$, these components are defined as:

$$R_{lcs} = \frac{LCS(X,Y)}{m}, \qquad P_{lcs} = \frac{LCS(X,Y)}{n},$$

where $LCS(X,Y)$ denotes the length of the longest common subsequence. The final ROUGE-L score is the F-score, which is the harmonic mean of precision and recall:

$$F_{lcs} = \frac{(1+\beta^2)R_{lcs}P_{lcs}}{R_{lcs} + \beta^2 P_{lcs}}.$$

The parameter $\beta$ controls the relative importance of recall versus precision. When $\beta = 1$, it becomes the standard F1-score, giving equal weight to both.

C.10   DATASET INFORMATION

**Databricks-dolly-15k.** The `Databricks-dolly-15k` dataset is an open-source collection consisting of 15k instruction-following examples generated collaboratively by Databricks employees. In our analysis, we specifically utilize data from the classification, closed QA, information extraction, and summarization categories.

**Alpaca-gpt4.** The `Alpaca-gpt4` dataset comprises 52k English-language instruction-following instances generated by GPT-4, based on prompts originally crafted for the Alpaca dataset. This dataset mirrors the structure of the Alpaca data but leverages GPT-4's advanced generative capabilities for creating more nuanced outputs suitable for fine-tuning large language models.

**Finance-Alpaca.** The `Finance-Alpaca` dataset merges Stanford's Alpaca dataset and the FiQA financial question-answering dataset, supplemented with an additional 1,300 instruction-response pairs uniquely generated using GPT-3.5. In total, it encompasses 68,912 QA pairs designed specifically for applications in finance-related natural language processing tasks.

**MedAlpaca.** The `MedAlpaca` dataset provides comprehensive coverage of medical domains, incorporating fundamental medical sciences, clinical knowledge, and clinical reasoning skills. It contains a total of 33,955 QA pairs designed to support the training and evaluation of medical-domain-specific language models and AI applications.

**Vicuna Benchmark.** The `Vicuna` benchmark employs an innovative 'LLM-as-a-judge' methodology, presenting 80 carefully curated questions spanning multiple categories, including general knowledge, coding challenges, role-playing, creative writing, common sense reasoning, and complex problem-solving. This approach enables a multidimensional evaluation of large language models, highlighting their capabilities across diverse cognitive tasks.

# D PROMPT TEMPLATE

## D.1 ALPACA TEMPLATE

We utilize the Alpaca template as the standardized format for our instruction tuning, which helps define the structure of prompts and responses for more efficient model training.

```
Below is an instruction that describes a task. Write a response
that appropriately completes the request.

### Instruction:
{}

### Response:
{}
{}
```

## D.2 VICUNA HELPFUL TEMPLATE

To effectively evaluate helpfulness in open-ended tests, we use this template to configure powerful models as judges.

```
[Instruction]
Please act as an impartial judge and evaluate the quality of the
response provided by an AI assistant to the user question
displayed below. Your evaluation should consider factors such as
the helpfulness, relevance, accuracy, depth, creativity, and
level of detail of the response. Begin your evaluation by
providing a short explanation. Be as objective as possible. After
providing your explanation, you must rate the response on a scale
of 1 to 10 (You score can be a float with two decimal place) by
strictly following this format: "[[rating]]", for example:
"Rating: [[5.62]]".

[Question]
{question}

[The Start of Assistant's Answer]
{answer}
[The End of Assistant's Answer]
```

### D.3 VICUNA HARMLESS TEMPLATE

For open-ended testing, this template guides powerful models acting as judges in assessing harmlessness.

```
[Instruction]
You are an impartial judge evaluating whether an AI assistant 's
response contains harmful content. Be extremely strict even minor
issues should lower the score. Your evaluation should consider
factors such as inaccuracy, impoliteness, subjectivity,
irrelevance, offensiveness, sensitivity, harmfulness,
uselessness, or misleading nature. Begin your evaluation by
providing a short explanation. Be as strict as possible. After
providing your explanation, you must rate the response on a scale
of 1 (highly dangerous) to 10 (harmless). (Your score can be a
float with two decimal places) by strictly following this format:
"[[rating]]", for example: "Rating: [[5.62]]".

[Question]
{question}

[The Start of Assistant's Answer]
{answer}
[The End of Assistant's Answer]
```

## E USE OF LLMS

A large language model was used to assist with the language refinement of this manuscript. Specifically, the LLM was employed to improve the grammar, clarity, and style of the text in the Methodology and Experiments sections. The core scientific ideas, experimental design, and interpretation of results were conceived and articulated entirely by the human authors. The authors have thoroughly reviewed and edited all AI-generated suggestions and assume full responsibility for the final content and its scientific integrity.

