# OpenReview forum: "FLEx: Personalized Federated Learning for Mixture-of-Experts LLMs via Expert Grafting"
_ICLR.cc/2026/Conference — ICLR 2026 Conference Withdrawn Submission_

### Official Review · Reviewer_KzTs · 2025-10-30

**Soundness:** 3
**Presentation:** 4
**Contribution:** 3
**Rating:** 6
**Confidence:** 5

**Summary:**

This paper introduces FLEx (Federated LLMs with Personalized Experts), a novel personalized Federated Learning (FL) framework for fine-tuning pre-trained Mixture-of-Experts (MoE) LLMs. FLEx addresses two major MoE-FL challenges—high communication cost and catastrophic forgetting—via a decoupled strategy:
1) Selective Aggregation: Only dense, shared non-expert parameters are aggregated, while all pre-trained expert parameters are kept frozen locally.
2) Expert Grafting: Each client performs a greedy selection to "graft" a lightweight personalized expert from the frozen pool, which is then jointly fine-tuned with a novel, trainable Adaptive Gating Mechanism. Experiments on non-IID instruction tuning show FLEx achieves superior personalization performance (ROUGE-L) and robust knowledge retention (MMLU) compared to FL baselines.

**Strengths:**

1) Effective Solution to MoE-FL: Successfully utilizes MoE sparsity to reduce communication overhead and leverage the frozen experts to prevent catastrophic forgetting.
2) Strong Performance Gains: Demonstrates significant improvements over standard FL baselines, particularly under challenging pathological non-IID settings.
3) Knowledge Preservation: MMLU results strongly support the efficacy of freezing experts for retaining general world knowledge.

**Weaknesses:**

1) Greedy Selection Limitation: The expert grafting strategy relies on a greedy selection process that picks only a single expert for personalization. While effective, this approach does not explore the potential gains—or added complexity—of using multiple experts or more sophisticated selection mechanisms.
2) Lack of Communication Cost Measurement: Although reduced communication is presented as a key advantage, the paper does not provide concrete empirical evidence—such as tables or numerical comparisons—to quantify how many fewer parameters or bytes are transmitted compared to a full MoE model under FedAvg.
3) Limited Generalizability: Experiments are conducted solely on one MoE architecture, Qwen1.5, and focus on NLP instruction tuning tasks. Results may not generalize to other MoE-based models or different modalities, and no cross-architecture or multi-domain validation is provided.

**Questions:**

1) The expert grafting uses single greedy selection. Did you investigate using a weighted combination or mixture of multiple experts for local adaptation? How would this compare in terms of performance vs. additional complexity or communication/computation cost?
2) Given that communication efficiency is a core contribution, could you provide a quantified analysis detailing the reduction ratio in transmitted parameters or total bytes per round compared to a MoE + FedAvg baseline?

---

### Official Review · Reviewer_679S · 2025-10-31

**Soundness:** 3
**Presentation:** 3
**Contribution:** 2
**Rating:** 4
**Confidence:** 4

**Summary:**

This paper proposes FLEx, a novel framework for federated learning with MoE-based LLMs. The central idea is to disentangle globally shared parameters from personalized components: only the dense non-expert parameters (e.g., attention layers) are aggregated across clients, while all pretrained experts remain locally frozen to retain general knowledge. To personalize the model, each client grafts a new lightweight expert by selecting the most effective frozen expert based on local data, and integrates it using a jointly trained adaptive gating mechanism. The proposed design aims to address both data heterogeneity and communication efficiency in FL scenarios.

**Strengths:**

- **Clarity and Structure:** The paper is clearly written and well-organized. Figures and algorithmic descriptions are intuitive, enhancing the accessibility of the methodological exposition.
- **Reproducibility:** The authors have provided code and implementation details, which supports reproducibility and validation of the reported results.

**Weaknesses:**

- **Potential Representation–Routing Misalignment:** The design updates dense non-expert layers through global aggregation while keeping all pretrained experts and their routers frozen. This raises concerns about potential misalignment between the evolving feature representations and the static routing mechanism, which could lead to suboptimal or unstable expert activation, especially under non-IID data distributions. The paper lacks analysis—such as expert utilization statistics or routing entropy—to evaluate this potential drift.
- **Unsubstantiated Efficiency Claims:** Although the method is described as efficient, inference actually involves routing each token to $K+1$ experts instead of $K$, introducing additional computational overhead. While the appendix suggests this overhead is negligible, no quantitative evidence (e.g., FLOPs, latency measurements) is provided to substantiate this claim.
- **Experimental Rigor:**
    - **Limited Baselines:** Experimental comparisons are confined to classical FL methods (e.g., FedAvg, FedProx, SCAFFOLD), omitting several recent personalized or MoE-based federated LLM approaches [1–4].
    - **Lack of Statistical Reporting:** Results appear to be based on single runs, with no reported variance or statistical significance tests, which limits the robustness of the empirical evaluation.

[1] Dual-personalizing adapter for federated foundation models, 2024.

[2] FDLoRA: personalized federated learning of large language model via dual LoRA tuning, 2024.

[3] FedMoE: Personalized Federated Learning via Heterogeneous Mixture of Experts, 2024.

[4] Personalized Federated Fine-Tuning for LLMs via Data-Driven Heterogeneous Model Architectures, 2024.

**Questions:**

Why does FLEx preserve all pretrained experts while adding one personalized expert per client? Would it not be more parameter-efficient to retain $N-1$ frozen experts and adapt one via a parameter-efficient method like LoRA, thereby maintaining the overall parameter budget while still enabling effective personalization?

---

### Official Review · Reviewer_Zkor · 2025-10-31

**Soundness:** 2
**Presentation:** 3
**Contribution:** 2
**Rating:** 2
**Confidence:** 4

**Summary:**

This paper addresses the significant challenges of applying federated learning (FL) to Mixture-of-Experts (MoE) large language models (LLMs), namely prohibitive communication costs, catastrophic forgetting of pretrained knowledge, and poor personalization under data heterogeneity. The authors propose FLEx (Federated LLMs with Personalized Experts), a novel framework that decouples shared knowledge from personalization. FLEx's core mechanism involves freezing all pretrained experts (preserving world knowledge and eliminating their communication cost) and aggregating only the shared, dense non-expert parameters (e.g., attention layers). To achieve personalization, FLEx introduces an "expert grafting" mechanism, where each client identifies the single, most suitable pretrained expert based on a local reconstruction loss. A copy of this expert is then added to the model as a new, trainable, client-specific expert. An adaptive gating mechanism is jointly trained to learn when to route to this new personalized expert versus the frozen pretrained ones. Experiments on non-IID instruction-following tasks show that FLEx outperforms standard federated baselines (like FedAvg) in average performance and demonstrates superior general knowledge retention as measured by MMLU.

**Strengths:**

S1. The paper identifies a critical and highly practical problem. The naive application of standard FL to massive MoE models is untenable due to communication overhead and knowledge corruption. The paper's core strategy--decoupling the aggregation of shared non-expert parameters from the personalization of frozen experts--is an elegant and effective solution to this problem.

S2. The "expert grafting" mechanism for personalization is a clever and efficient approach. Instead of training a new, randomly initialized expert from scratch (which would be data-hungry and slow to converge), the framework leverages the rich, pretrained knowledge by copying the most relevant existing expert. This provides a high-quality initialization for the personalized expert, allowing for rapid and effective adaptation to local data.

**Weaknesses:**

W1. The framework's core design--aggregating only non-expert parameters while freezing all pretrained experts--prevents the model from collaboratively learning new, shared knowledge within its most critical components. In Transformer architectures, the expert layers (FFNs) are the primary location for knowledge storage, whereas the aggregated attention layers mainly handle information routing. By freezing all experts, the FL process is blocked from updating the model's core "knowledge" stores with insights from multi-client data. This is a major trade-off. A more robust solution might involve a compromise, such as including one or more shared experts that are aggregated by the server like DeepSeekMoE, allowing the model to collaboratively learn new shared concepts without the full cost of aggregating all experts. The paper should investigate this possibility.

W2. The core "expert grafting" mechanism is limited to selecting only a single expert for personalization (per layer). The paper's own ablation study (Appendix C.2, Table 7) shows that performance improves (albeit slightly) when grafting 2 or 3 experts, though this is dismissed due to the computational cost of the selection process. This suggests the "one expert" design is a heuristic for efficiency, not necessarily the optimal choice for personalization. For clients with genuinely multi-modal data (e.g., a mix of finance and medical texts), a single personalized expert may be forced to compromise, whereas a small set of personalized experts would be a more natural fit.

W3. The paper's baseline comparisons are not as strong as they could be. The baselines (FedAvg, FedProx, etc.) are all applied in a "naive" full-aggregation setting, which the paper itself argues is flawed and non-viable. A more meaningful comparison would be against other parameter-efficient federated learning (PEFT-FL) strategies. For example, how does FLEx (which trains a full expert copy + gates) compare to a standard Federated LoRA (FLoRA) applied to all parameters (including experts), or FLoRA applied only to the non-expert layers? Without these comparisons, it's difficult to ascertain if FLEx's specific "grafting" design is superior to other, more established PEFT-FL techniques.

W4. The performance improvements in some key experiments are marginal and lack statistical validation. For instance, in Table 2 ($\alpha=1.0$), the improvement of FLEx (37.54) over the FedAvg baseline (37.51) is less than 0.1 ROUGE-L points. Similarly, the gains in Helpfulness and Harmlessness (Figure 3) over the next-best methods are small. The paper does not report variance or standard deviation for these results, making it impossible to assess whether these minor differences are statistically significant or simply noise from a single experimental run.

**Comments**

C1. As a point of clarification on Table 1: In the "Summ" (Summarization) column, the paper's markings are consistent with the caption. The best score is MoE + FedAvgM (42.54) and the second-best is MoE + FedAdagrad (42.24), which are bolded and underlined respectively. The FLEx score (42.09) is third in this specific category, which should not be underlined.

**Questions:**

See weakness

---

### Official Review · Reviewer_kmrK · 2025-11-01

**Soundness:** 2
**Presentation:** 2
**Contribution:** 2
**Rating:** 2
**Confidence:** 4

**Summary:**

This paper proposes FLEx, a federated learning framework for Mixture-of-Experts (MoE) models. To reduce communication cost and avoid catastrophic forgetting, FLEx aggregates only the dense non-expert parameters while keeping pretrained experts frozen. To handle data heterogeneity, it introduces an expert grafting mechanism that builds lightweight personalized experts via pruning and integrates them through an adaptive gating module. Experiments on non-IID instruction-tuning datasets reportedly show improved personalization and preserved general knowledge compared with existing federated baselines.

**Strengths:**

1. The paper tackles an important and underexplored problem of enabling personalization for large MoE-based LLMs in federated settings.

2. The motivation (preserving pretrained knowledge and avoiding high communication cost) is sound and practically relevant.

3. The overall framework is well-written and supported with some empirical evaluation.

**Weaknesses:**

1. The term “expert grafting” is semantically misleading. In biology, grafting refers to physically attaching a part of one organism onto another to form an integrated system, such as the work FedGraft [R1]. In this work, however, the method only selects or prunes pretrained experts and attaches a small adapter for local personalization. This is conceptually closer to adapter-based personalization or lightweight expert extension, rather than genuine “grafting”.
Using an inaccurate metaphor may cause conceptual confusion and oversell the contribution. The authors should revise the terminology to reflect the actual mechanism.

2. Despite the new terminology, the proposed mechanism closely resembles prior personalized adapter or LoRA-style methods, such as [R2-R4], where clients retain frozen pretrained parameters and inject small local modules for adaptation. The authors need to compare and discuss with relevant baselines.

3. Although the paper claims to address personalized federated learning (PFL), it does not compare the proposed method with any state-of-the-art personalized FL baselines or recent MoE-based FL methods, such as [R5-R8]. This omission severely limits the credibility of the claimed effectiveness. Without such comparisons, it is difficult to assess whether the proposed approach truly advances the SOTA.

4. The proposed method shows limited performance improvement and even underperforms some baselines under non-IID settings.

5. The evaluation focuses on one dataset (Databricks-Dolly-15K) and one model (Qwen1.5-MoE-A2.7B). There is no analysis of scalability (datasets, model, and different numbers of clients) and ablation on pruning thresholds.

6. Lacks of convergence analysis.

[R1] FedGraft: Memory-Aware Heterogeneous Federated Learning via Model Grafting

[R2] FLoRA: Federated Fine-Tuning Large Language Models with Heterogeneous Low-Rank Adaptations

[R3] Federated Fine-tuning of Large Language Models under Heterogeneous Tasks and Client Resources

[R4] Selective Aggregation for Low-Rank Adaptation in Federated Learning

[R5] GPFL: Simultaneously Learning Global and Personalized Feature Information for Personalized Federated Learning

[R6] Exploiting Shared Representations for Personalized Federated Learning

[R7] FedMoE: Personalized Federated Learning via Heterogeneous Mixture of Experts

[R8] FedMoE-DA: Federated Mixture of Experts via Domain Aware Fine-grained Aggregation

**Questions:**

Please see weaknesses.

---

### Note · Authors · 2025-11-12

I have read and agree with the venue's withdrawal policy on behalf of myself and my co-authors.